# Protective Factors and Coping Styles Associated with Quality of Life during the COVID-19 Pandemic: A Comparison of Hospital or Care Institution and Private Practice Nurses

**DOI:** 10.3390/ijerph19127112

**Published:** 2022-06-10

**Authors:** Jonathan Jubin, Philippe Delmas, Ingrid Gilles, Annie Oulevey Bachmann, Claudia Ortoleva Bucher

**Affiliations:** 1La Source School of Nursing, HES-SO University of Applied Sciences and Arts Western Switzerland, Av. Vinet 30, 1004 Lausanne, Switzerland; p.delmas@ecolelasource.ch (P.D.); a.oulevey@ecolelasource.ch (A.O.B.); c.ortolevabucher@ecolelasource.ch (C.O.B.); 2Epidemiology and Health Systems, Center for Primary Care and Public Health, 1010 Lausanne, Switzerland; ingrid.gilles@unisante.ch

**Keywords:** nurses, quality of life, protective factors, coping strategies, type of practice, COVID-19

## Abstract

In France, nurses work either in hospitals and care institutions or in private practice, following physicians’ prescriptions and taking care of patients at their homes. During the COVID-19 pandemic, these populations of nurses were exposed to numerous sources of stress. The main objective of the present study was to identify the protective factors they mobilized to face the crisis and how these factors contributed to sustaining their quality of life (QoL). A cross-sectional study was conducted to answer these questions. Overall, 9898 French nurses participated in the study, providing demographic information and filling out QoL (WHOQOL-BREF), perceived stress (PSS-14), resilience (CD-RISC), social support (MSPSS), and coping style (BRIEF-COPE) questionnaires. The results revealed very few differences between the two groups of nurses, which is surprising given the drastically different contexts in which they practice. Social support and two coping strategies (positive reframing and acceptance) were associated with a high QoL, whereas perceived stress and four coping strategies (denial, blaming self, substance use, and behavioral disengagement) were associated with poor QoL. In the light of these results, we recommended promoting social support and coping strategies to help nurses cope during the pandemic.

## 1. Introduction

The COVID-19 pandemic appeared in China in the fall of 2019 and has since spread worldwide [1]. In February 2021, when the third wave occurred, the virus had infected hundreds of millions of people and millions had died [2,3]. Moreover, numerous severe cases led to hospitalizations, putting the healthcare system and professionals under heavy pressure over a prolonged period of time [4].

Nurses have been among the most sought-after healthcare workers during the pandemic. During the different waves of COVID-19, they faced death, uncertainty, and work overload every day [4]. Such circumstances are likely to elicit stress, depressive symptoms, insomnia, and anxiety [5,6]. In turn, stress can lead to cognitive overload that can result in errors at work [7,8] and feelings of a lower quality of life (QoL), which is a strong correlate of nurses’ engagement in dire contexts [9,10,11]. QoL is defined by the World Health Organization as ‘an individual’s perception of their position in life in the context of the culture and value systems in which they live and in relation to their goals, expectations, standards, and concerns [12]. This definition highlights the multi-dimensionality of the QoL concept that articulates four broad domains: physical health, psychological wellbeing, social relationships, and environment. During the COVID-19 crisis, QoL was found to be associated with the use of adaptive strategies, particularly, resilience and coping [13,14]. QoL is thus a key indicator of an individual’s well-being or life experiences during crises and it allows us to understand how nurses adapt to crises [15].

Nurses’ QoL has been extensively explored during the COVID-19 pandemic [16]. However, most researchers have adopted a ‘pathogenic’ perspective [17], focusing on risk factors that impair health [18,19]. Such a pathogenic approach is limited because it tends to categorize individuals as sick or healthy and considers that disease prevention consists solely in avoiding or minimizing risk factors [20]. Hence, this approach does not necessarily provide nurses with tools to anticipate or cope with crises such as a pandemic. Thus, identifying protective factors seems to be a more efficient strategy to allow crisis anticipation and provide nurses with actionable resources, particularly when the sources of stress cannot be avoided. Thus, in the present study, we adopted a salutogenic approach, which aims at identifying factors that protect one’s QoL and health [17]. In the context of the COVID-19 pandemic and its duration, investigating which protective factors might reinforce nurses’ ability to deal with stress is of paramount importance. Salutogenic-oriented studies could help us understand how to maintain or improve nurses’ QoL and health. Protective factors are relevant to guiding the development of interventions aimed at educating nurses to help them cope with prolonged a crisis [17,21,22].

Nevertheless, investigators have taken little interest in identifying protective factors. We identified three key resources to help nurses cope with stress during the pandemic. First is resilience, which reflects the ability of an individual to face the difficulties of life and bounce back after a traumatic event [23,24]. Resilience has been found to mediate the association between sources of stress and psychological distress during COVID-19, and several studies recommend training nurses to become resilient to prepare them for crises [5,25,26]. The second is social support, which measures the psychological and material support provided by an individual’s entourage [27,28]. Lastly, the third, is coping strategies that individuals use to react and adapt to stressful situations [29]. Maladaptive coping during the pandemic has been shown to be associated with poor outcomes [6]. On the other hand, several studies point to the conjoint effect of resilience, social support, and coping strategies [26]. For example, resilience combined with other adaptive coping strategies has been shown to help manage stress and sustain personal growth [30]. To our knowledge, no study has explored the role of these factors in maintaining nurses’ QoL. Therefore, the main objective of this study was to identify which protective factors were most helpful in maintaining the nurses’ QoL during the pandemic in France.

Studying the protective role of resilience, social support, and coping strategies is important in France because nurses can work as employees at hospitals or care institutions (NHCIs) (e.g., retirement homes, care homes) or have a private independent practice (NPPs) where they provide care prescribed by a physician at patients’ homes [31]. Both categories of nurses had to adapt to multiple sources of stress during the pandemic. A survey of French NPPs in 2020 found that around a third of the respondents did not have sufficient medical supplies to do their work, and half of them reported having to reorganize their work schedules and feeling stressed by their work during the pandemic [32]. However, to our knowledge, no study has investigated the similarities and differences in how these two groups reacted to the pandemic. Such knowledge would allow us to provide nurses with the necessary support to help them adapt to the specific sources of stress in their practice during crisis times. Thus, the second objective of the current study was to investigate the role of different protective factors in determining QoL specific to the two types of practice (NHCI and NPP) and across both groups.

## 2. Materials and Methods

### 2.1. Design and Population

This cross-sectional survey was conducted in France between February and March 2021 during the third COVID-19 wave. An invitation to participate in the study accompanied by a link to the online self-administered questionnaire was sent to all registered nurses through the Ordre National des Infirmiers (French National Order of Nurses), followed by three weekly reminders. About 400,000 nurses are registered at the Ordre National des Infirmiers. Participation was anonymous and voluntary. The questionnaire was developed with Sphinx iQ2 v7.4.5.1. The data used in this study were part of a larger study focused on nurses’ QoL and health-protecting factors in European countries [33].

### 2.2. Measures

All instruments used in the larger European study have been previously validated in English and French. The instruments were administered in French.

*World Health Organization Quality of Life brief version (WHOQOL-BREF)* [34]. It includes 26 items measuring physical, psychological, social, and environmental aspects of QoL using 5-points Likert scales. It also includes two items measuring overall QoL and state of health, respectively. The questionnaire, as well as its French translation, have proven to be reliable and valid [34,35]. As recommended by the authors of the questionnaire, mean scores were transformed to range from 0 (poor QoL) to 100 (good QoL). As we were interested in measuring QoL as a global construct, we used the mean score of all items. Cronbach’s α revealed good internal consistency (.91) for the total QoL score.

*Perceived Stress Scale (PSS-14)* [36]. The French translation of this scale has been shown to have good psychometric properties [37]. It includes 14 items rated from 1 (low perceived stress) to 5 (high perceived stress) that we averaged to compute the perceived stress score. Cronbach’s α revealed good internal consistency (.90) for the total stress score.

*Connor-Davidson Resilience Scale (CD-RISC)* [24]. It contains 10 items rated from 1 (low resilience) to 5 (high resilience) and has been translated and validated in French [38]. Cronbach’s α revealed good internal consistency (.88) for the total resilience score.

*Multidimensional Scale of Perceived Social Support (MSPSS)* [28]. It measures perceived social support from family, friends, and significant other (i.e., any one person to whom the individual feels especially close) and consists of 12 items rated from 1 (low social support) to 7 (high support). The French translation exhibited good psychometric properties [39]. Cronbach’s α revealed good internal consistency (.94) for the total social support score.

*Coping Orientation to Problems Experienced Inventory (Brief-COPE)* [40]. This scale measures individuals’ favored coping strategies. It includes 14 dimensions, each measured by two items, that represent a coping strategy. The items are rated from 1 (rare use of that coping strategy) to 4 (frequent use). We calculated the mean of the two items for each dimension, and as Cronbach’s alpha may underestimate the reliability of two-items scales, we evaluated their reliability by using the Spearman-Brown coefficient rs as recommended by Eisinga et al. [41].

The 14 dimensions and their brief descriptions are as follows: Active coping refers to individuals actively attempting to suppress their problems or their effects (rs = 0.56); Planning consists of devising steps to best manage problems (rs = 0.69); Seeking instrumental support refers to seeking advice or help (rs = 0.78); Seeking emotional support refers to seeking moral support or sympathy (rs = 0.73); Venting refers to expressing emotions about problems (rs = 0.71); Positive reframing involves reassessing problem situations as positive (rs = 0.77); Acceptance is acknowledging the existence of problems (rs = 0.69); Denial is refusing to acknowledge the existence of problems (rs = 0.60); Self-blame is reproaching oneself for problems (rs = 0.56); Humor is not taking problems seriously (rs = 0.75); Religion is seeking solace in religious beliefs (rs = 0.84); Self-distraction is diverting one’s attention away from problems by focusing on something else (rs = 0.36); Substance use is escaping reality by consuming alcohol or drugs (rs = 0.94); Behavioral disengagement is abandoning goals prevented by problems (rs = 0.62) [42]. The French translation showed acceptable psychometric properties [42].

*Sociodemographic Variables*. Participants were asked to indicate their gender (male, female, ‘I define myself otherwise’), age category (18–29 years old, 30–39, 40–49, and 50 or more), marital status (married, single, other), and having children (yes, no). The question on ‘marital status other’ was open-ended, and most who responded indicated they were divorced (73.3%). Other questions asked were as follows: how long they have had their nursing diploma (less than 5 years, 5 to 10 years, or more than 10 years); if they had been reassigned to another service other than their usual one during the pandemic (yes, no); if at any point, they had been exposed to COVID-19 during their work (direct exposure: worked in a COVID-specific unit; indirect exposure: worked in a non-COVID-specific unit but that they received some COVID patients; and no exposure: no COVID patient was admitted to the unit); and whether they were engaged in private practice (NPPs) or a hospital/institutional care setting (NHCIs).

### 2.3. Data Analysis

All variables were treated as continuous except for sociodemographic variables that were treated either as dichotomized variables or as dummy variables (more than two categories). Descriptive analyses were first used to describe the sample and independent samples *t*-tests and Chi-square tests of independence were used to compare the two groups of nurses (NHCIs vs. NPPs). We also computed Pearson’s correlations and, after checking for linearity and normality assumptions, we conducted multiple linear regressions with QoL as the outcome variable: on (a) the full sample, and (b) stratified by the type of practice samples. We checked for multicollinearity among the predictive variables using the variance inflation factor (VIF) index, which revealed no problematic collinearity (all VIFs < 3) among the predictor variables [43]. As our sample was very large, we used listwise deletion for handling missing values and we lowered the significance threshold to 0.005 to minimize type I error [44]. All analyses were performed using R 4.1.1.

## 3. Results

### 3.1. Descriptive Characteristics

A total of 9898 nurses completed the questionnaire, with a response rate of about 2.5%. Participants’ characteristics are shown in Table 1. Overall, 85.1% of respondents were women, 14.0% were 18 to 29 years old, 26.4% were 30 to 39, 30.7% were 40 to 49, and 28.1% were 50 and more. Concerning work practice, 55.4% were NHCIs, 39.0% were NPPs, and 5.6% were both. This last group was included in the general analyses but, because of its small size (*n* = 553) compared to the others, it was not included in the stratified analyses.

Moreover, 26.7% of the respondents had worked in COVID-19-specific units, 50.1% had worked in facilities not initially dedicated to COVID-19 but that were temporarily transformed to receive COVID-19 patients, and 20.0% had only worked in units that never received COVID-19 patients. Finally, 25.2% of the participants had been reassigned to another service than their usual one at least once since the beginning of the pandemic.

Most coping strategies were reported to be used almost equally often (mean scores ranged between 2.2 and 2.7; SD ranged from 0.67 to 0.78), except for denial, humor, religion, substance use, and behavioral disengagement, which participants used less often (means ranged between 1.3 and 1.9 and SDs ranged between 0.59 and 0.80).

### 3.2. Analyses

The Chi-square test of independence (Table 1) conducted on the two samples revealed that NHCIs were generally younger (χ^2^(3) = 538.75, *p* < 0.001) and had had their diploma for a shorter time than NPPs had (χ^2^(2) = 753.33, *p* < 0.001). NHCI were also less frequently exposed indirectly to COVID-19, but more frequently exposed directly (χ^2^(2) = 98.85, *p* < 0.001).

Independent sample *t*-tests showed that NHCIs tended to receive more social support (t = 6.07, *p* < 0.001, *d* = 0.13) and were less resilient (t = −6.60, *p* < 0.001, *d* = 0.14) than NPPs, though effect sizes were very small. Moreover, the NHCIs reported seeking instrumental (t = 7.53, *p* < 0.001, *d* = 0.16) and emotional (t = 7.95, *p* < 0.001, *d* = 0.17) support, venting (t = 4.71, *p* < 0.001, *d* = 0.10), turning to religion (t = 4.85, *p* < 0.001, *d* = 0.10), and exhibiting behavioral disengagement (t = 5.58, *p* < 0.001, *d* = 0.12) more often than NPPs. On the contrary, NHCIs used active coping (t = −3.57, *p* < 0.001, *d* = 0.08), planning (t = −3.58, *p* < 0.001, *d* = 0.08), positive reframing (t = −5.85, *p* < 0.001, *d* = 0.12), and acceptance (t = −4.46, *p* < 0.001, *d* = 0.09) less often than NPPs.

Pearson’s correlations among the variables are provided in Table 2. The strongest correlations were between QoL and perceived stress (*r* = −0.69), planning and active coping (*r* = 0.65), seeking instrumental and emotional support (*r* = 0.61), and seeking instrumental support and venting (*r* = 0.60). All other rs were below 0.60. With a sample size as large as 9898, any correlation larger than 0.04 would be significant at *p* < 0.0001. It is thus not surprising that almost all correlations were statistically significant. Consequently, the correlations were not interpreted based on obtained *p*-values alone.

### 3.3. Regression Analysis

We performed multiple linear regression analyses, first on the full sample and then on each subsample (type of practice). In each analysis, QoL was the outcome variable, and the predictors were perceived stress, social support, resilience, coping styles, exposure to COVID-19, reassignment during the pandemic, and sociodemographic variables (Table 3). Adjusted R^2^ values ranged from 0.62 to 0.63 in the three analyses.

The analyses revealed that perceived stress was associated with QoL in the full sample (β = −0.49, 95% CI: [−0.51, −0.48]), and in the NHCI and NPPs samples (β’s = −0.49 and −0.50, respectively): Respondents reporting greater perceived stress also reported lower quality of life. Social support was positively associated with QoL in all analyses (β’s ranged between 0.20 and 0.21, *p* < 0.005): the more respondents reported having social support, the higher their QoL. Lastly, resilience was not significantly associated with QoL (β’s ranged between 0.01 and 0.02, *p* > 0.005).

Higher level of coping-oriented venting (β’s ranged between 0.03 and 0.05, *p* < 0.005 in the three analyses), as well as active coping (β’s ranged between 0.01 [NHCI] and 0.04 [full sample], *p* < 0.005, positive reframing (β’s ranged between 0.08 and 0.12, *p* < 0.005 in the three analyses, and acceptance (β’s ranged between 0.06 and 0.08, *p* < 0.005), were associated with higher levels of QoL. Higher levels of coping oriented toward denial (β’s ranged between −0.04 and −0.03, *p* < 0.005), blaming one-self (β’s ranged between −0.08 and −0.06, *p* < 0.005), substance use (β’s ranged between −0.04 and −0.03, *p* < 0.005), and behavioral disengagement (β’s ranged between −0.05 and −0.06, *p* < 0.005 were associated with lower levels of QoL.

Direct and indirect exposures to COVID-19 compared with no exposure were not associated with QoL (β’s ranged between −0.03 and −0.01, *p* > 0.005). Neither was reassignment during the pandemic (β’s ranged between −0.01 and 0.00, *p* > 0.005). Age was negatively associated with QoL, with older participants reporting lower QoL (β’s ranged between −0.07 to −0.10 for age classes compared to 18–29 years old, *p* < 0.005 for NHCI and the full sample).

## 4. Discussion

In France, nurses work in hospitals or care institutions or have private practices [31]. Both groups of practitioners experienced many changes during the COVID-19 pandemic [4]. However, to our knowledge, no study has investigated which protective factors were mobilized by nurses in these different contexts to preserve their QoL during the pandemic. Thus, the goals of the present study were to identify protective factors used by nurses and assess if the protective factors used differed as a function of their type of practice (NHCI or NPP). The results suggest that, for both groups of nurses, high levels of perceived stress and problem-avoidant coping strategies were associated with poor QoL, whereas high social support and solution-oriented coping strategies were associated with good QoL. Surprisingly, resilience was not significantly associated with QoL in either practice.

Nurses’ stress has been investigated in many studies [45]. Researchers have shown that even in ordinary times (i.e., without a pandemic) nurses face many sources of stress resulting from their workload or patient-related issues, which affects their QoL [11,46,47]. The COVID-19 pandemic has exacerbated occupational stress because of the pressure put on care services and the central role nurses play in these services [48]. Prior research has shown that social support played an important role in protecting nurses from negative outcomes (e.g., burnout) caused by their exposure to multiple stress sources and helped them to preserve their QoL during the pandemic [49,50]. Our results confirmed these findings. The stress caused by high levels of uncertainty during the current crises may have increased the need for social support [51]. Undoubtedly, social support (providing assistance and information as needed) allows nurses to cope with both feelings of uncertainty and perceived stress [52,53]. The provision of social support at work can be problematic for NPPs because they cannot receive direct support from their institutions or colleagues. Their professional interactions are restricted to patients, their families, and the patients’ physician [31], thus limiting the possibility of sharing experiences and asking for advice. On the contrary, NHCIs, who are included in teams, have more possibilities to receive support in the regular course of their work compared to NPPs. Our results support this assumption since NHCIs reported higher scores on social support than NPPs, though the effect size was very small. Nonetheless, the protective factors associated with QoL differed very little between the two types of nurses despite drastic differences in the nature of their practice. To our knowledge, no study has specifically compared NHCIs and NPPs on determinants of QoL. It is thus difficult to explain that, despite the difference in perception of social support, perception of QoL did not differ between nurses working in these different practice contexts. Nurses may have adapted differently in each work setting to the pandemic by relying on coping strategies appropriate to their own setting to maintain their QoL.

Our analyses confirmed the importance of coping strategies to preserve QoL, which is consistent with findings from previous research [29]. However, our study showed that NHCIs more frequently used coping strategies involving social relationships, such as seeking emotional and instrumental support or venting. NPPs used more individualistic coping strategies, such as active coping, planning, positive reframing, and acceptance. This indicates that both groups of nurses used some strategies to protect their QoL, but these strategies differed according to the professional context. This is what was proposed by Moos in his conceptual framework linking context and coping [54]. To him, cognitive appraisal of the stressful situation and coping responses are interdependent with, among others, an individual’s environmental system (i.e., supra-personal and social climate factors, such as pressures arising from professional activities that threaten health). Although developed to understand coping patterns in everyday life, this observation could be valid in professional settings where individuals are exposed to numerous stress sources. Moreover, it is noteworthy that the two coping strategies that were the most strongly associated with a satisfying QoL were positive reframing and acceptance. These strategies focus on finding the positive aspects of stressful situations while still acknowledging the existence of negative events [55,56]. Focusing on positive affect generates positive emotions that, according to Fredrickson [57], lead to enhancing one’s social, intellectual, and physical resources, all of which are durable and can be mobilized in other stressful situations. Hence, developing and encouraging the use of these strategies could help increase nurses’ QoL. Additionally, negative coping strategies such as denial, blaming self, substance use, and behavioral disengagement were associated with low QoL. The use of these strategies reduces negative affect generated by stressful situations without addressing them [58]. Thus, the continued use of negative coping strategies is likely to lead to an accumulation of problems that might eventually seem insurmountable, which in turn, can elicit more avoidant behaviors, creating a vicious circle [14,59]. Our finding that both NHCIs and NPPs seldom used these strategies indicates that most of them adapted well to their contexts by using healthy coping strategies.

Surprisingly, resilience was not significantly associated with QoL, a finding contrary to our expectations based on previous research [14,23,26]. It is possible that the inclusion of the coping strategies in the analyses masked the link between resilience and QoL since resilience and coping tend to be closely associated [60,61]. The correlations we observed between resilience and some coping strategies, such as active coping, planning, positive reframing, acceptance, and humor, support this hypothesis. To assess this, a mediation analysis should be conducted. However, the present design using the Brief-COPE inventory, which assesses 14 coping strategies, does not lend itself well to such analysis. Another explanation could be that resilience has an indirect effect on QoL. Indeed, the relationship between resilience and QoL is not precisely defined [62], and QoL is a broad concept that includes both physical and psychological aspects [63]. Thus, resilience might not affect QoL when considered as a composite concept, but it might moderate the effect that crises may have on some of the facets of QoL.

QoL can be improved by working on strengthening social support and promoting problem-oriented coping strategies while discouraging problem-avoidant behaviors. Given the similarity of the associations between these variables and QoL for NHCIs and NPPs, such strategies could benefit both groups. Our findings support the development of primary prevention interventions aimed at all nurses. They also advocate applying salutogenic models in the healthcare system, such as Neuman’s Systems Model, which considers protective factors and depicts their role as buffers against the stressors that individuals encounter [64,65]. Lastly, our results support the use of management policies and practices that foster social support to encourage and help nurses face the sources of stress they encounter rather than avoid them.

The main limitation of the present study was its cross-sectional nature, which did not allow for causal explanations. Moreover, some of the differences and regression coefficients we reported were of small magnitude, although statistically significant. Furthermore, NPPs were older on average and more experienced than NHCIs because, to work in private practice, nurses must have worked for at least two years at a hospital or care institution. Additionally, NPPs do not exist in all countries or might take different forms depending on local culture and legislation. Comparisons between the present study results and those found in other countries with different healthcare systems and organizations thus might be limited. As this study used self-reported measures and volunteer samples, our findings do not generalize to reflect the characteristics of non-respondents. Finally, the present study only investigated individual protective factors, but organizational and environmental factors might also play an important role in nurses’ QoL [66,67]. It should not be inferred from the results of our study that nurses are solely responsible for protecting themselves from professional stress sources via the use of protective factors and coping strategies.

## 5. Conclusions

When exposure to sources of stress is unavoidable, individuals have no choice but to rely on protective factors and coping strategies to maintain their QoL. The present study showed that some of these factors and strategies were indeed strongly associated with QoL in a sample of French nurses during the COVID-19 pandemic, whether they worked at hospitals or care institutions or in private practice. Interventions targeted to help nurses seek more social support, and use acceptance and positive reframing strategies could help preserve or increase their QoL. Based on our results, experimental studies might be developed to assess the causality of the observed associations. Additionally, the identification of nurses’ QoL protective factors should be expanded to organizational or environmental factors, other countries, and other domains of practice to develop recommendations specific to the context of nursing practice.

After the COVID-19 pandemic, the future will likely hold new challenges, such as natural disasters and new diseases. Again, nurses will have to face these challenges on the frontline. It is thus of paramount importance that measures are taken to help them protect their QoL during stressful events.

## Figures and Tables

**Table 1 ijerph-19-07112-t001:** Descriptive characteristics for the whole sample and both types of practice.

	All (*n* = 9898)	NHCI (*n* = 5485)	NPP (*n* = 3860)	*p*-Values	
**Sociodemographic variables**	Frequency	Frequency	Frequency		
Gender: Men	14.7%	14.0%	15.5%	0.136	
Gender: Women	85.1%	85.7%	84.3%		
Gender: Describes otherwise	0.1%	0.1%	0.1%		
Age: 18–29	14.0%	20.2%	4.5%	<0.001	
Age: 30–39	26.4%	27.7%	25.0%		
Age: 40–49	30.7%	26.9%	36.7%		
Age: 50 or greater	28.1%	24.6%	32.9%		
Marital situation: Single	20.1%	21.5%	17.1%	<0.001	
Marital situation: Married	76.0%	74.8%	78.8%		
Marital situation: Other	3.7%	3.5%	3.9%		
Having Children: Yes	70.1%	63.6%	80.6%	<0.001	
Time since diploma: less than 5 years	14.3%	21.3%	3.5%	<0.001	
Time since diploma: 5–10 years	19.1%	21.6%	15.3%		
Time since diploma: more than 10 years	66.3%	56.8%	80.9%		
**COVID-19-related variables**	Frequency	Frequency	Frequency		
Exposure: None	20.0%	23.5%	14.9%	<0.001	
Exposure: Indirect	50.1%	48.1%	53.4%		
Exposure: Direct	26.7%	28.0%	23.9%		
Reassignment: Yes	25.2%	33.2%	11.9%	<0.001	
**Main independent variables**	Mean (SD)	Mean (SD)	Mean (SD)		Cohen’s *d*
Quality of Life	59.5 (14.8)	59.9 (14.7)	59.0 (15.0)	0.006	0.06
Perceived Stress	3.1 (0.6)	3.1 (0.6)	3.1 (0.6)	0.711	0.01
Social Support	5.4 (1.2)	5.4 (1.2)	5.3 (1.3)	<0.001	0.13
Resilience	3.6 (0.7)	3.5 (0.7)	3.6 (0.7)	<0.001	0.14
**Copings styles variables**	Mean (SD)	Mean (SD)	Mean (SD)		
Active Coping	2.72 (0.67)	2.70 (0.67)	2.75 (0.67)	<0.001	0.08
Planning	2.67 (0.73)	2.65 (0.72)	2.71 (0.73)	<0.001	0.08
Seeking Instrumental Support	2.35 (0.77)	2.40 (0.76)	2.28 (0.78)	<0.001	0.16
Seeking Emotional Support	2.35 (0.75)	2.40 (0.75)	2.27 (0.75)	<0.001	0.17
Venting	2.46 (0.78)	2.49 (0.76)	2.41 (0.79)	<0.001	0.10
Positive Reframing	2.70 (0.77)	2.66 (0.77)	2.75 (0.78)	<0.001	0.12
Acceptance	2.62 (0.75)	2.59 (0.74)	2.66 (0.75)	<0.001	0.09
Denial	1.49 (0.63)	1.48 (0.62)	1.51 (0.64)	0.048	0.04
Self-Blame	2.24 (0.67)	2.25 (0.68)	2.22 (0.67)	0.038	0.04
Humor	1.91 (0.76)	1.89 (0.76)	1.94 (0.76)	0.002	0.07
Religion	1.53 (0.80)	1.56 (0.81)	1.48 (0.76)	<0.001	0.10
Self-distraction	2.62 (0.70)	2.63 (0.69)	2.59 (0.71)	0.009	0.06
Substance Use	1.36 (0.63)	1.34 (0.61)	1.37 (0.63)	0.027	0.05
Behavioral Disengagement	1.46 (0.59)	1.49 (0.61)	1.43 (0.57)	<0.001	0.12

Remark: some nurses who participated in the study had a mixed practice alternating between hospital/care institution and private work. They were included in the general analyses but, because of how few they were in number (*n* = 553) compared to the other two categories, they were not included in the stratified analyses. NHCI: nurses working at hospitals or care institutions; NPP: nurses in private practice.

**Table 2 ijerph-19-07112-t002:** Pearson’s correlations among all variables for the full sample.

	1.	2.	3.	4.	5.	6.	7.	8.	9.	10.	11.	12.	13.	14.	15.	16.	17.	18.
1. Quality of Life	-																	
2. Perceived Stress	**−0.69**	-																
3. Social Support	**0.44**	**−0.22**	-															
4. Resilience	**0.45**	**−0.43**	**0.24**	-														
5. Active Coping	**0.35**	**−0.27**	**0.20**	**0.48**	-													
6. Planning	**0.30**	**−0.20**	**0.20**	**0.41**	**0.65**	-												
7. Seeking Instrumental Support	**0.17**	0.01	**0.39**	**0.05**	**0.23**	**0.24**	-											
8. Seeking Emotional Support	**−0.06**	**0.21**	**0.23**	**−0.14**	**0.06**	**0.11**	**0.61**	-										
9. Venting	**0.26**	**−0.09**	**0.38**	**0.19**	**0.29**	**0.27**	**0.60**	**0.39**	-									
10. Positive Reframing	**0.50**	**−0.42**	**0.26**	**0.56**	**0.46**	**0.45**	**0.17**	**−0.03**	**0.25**	-								
11. Acceptance	**0.44**	**−0.40**	**0.18**	**0.44**	**0.39**	**0.38**	**0.09**	**−0.09**	**0.19**	**0.52**	-							
12. Denial	**−0.27**	**0.26**	**−0.10**	**−0.15**	**−0.07**	**−0.08**	0.01	**0.12**	**−0.07**	**−0.15**	**−0.24**	-						
13. Self-Blame	**−0.30**	**0.30**	**−0.13**	**−0.26**	**−0.06**	0.00	**0.10**	**0.23**	−0.01	**−0.20**	**−0.11**	**0.20**	-					
14. Humor	**0.35**	**−0.35**	**0.17**	**0.46**	**0.26**	**0.25**	**0.08**	**−0.08**	**0.16**	**0.48**	**0.37**	**−0.10**	**−0.10**	-				
15. Religion	**0.06**	0.00	**0.05**	**0.06**	**0.11**	**0.10**	**0.12**	**0.09**	**0.12**	**0.17**	**0.08**	**0.05**	−0.01	**0.05**	-			
16. Self-Distraction	**0.05**	0.00	0.02	**0.13**	**0.20**	**0.15**	**0.14**	**0.17**	**0.12**	**0.14**	**0.13**	**0.07**	**0.15**	**0.10**	**0.10**	-		
17. Substance Use	**−0.25**	**0.23**	**−0.13**	**−0.15**	**−0.12**	**−0.12**	**−0.05**	**0.09**	**−0.06**	**−0.17**	**−0.15**	**0.14**	**0.15**	**−0.06**	−0.02	**0.04**	-	
18. Behavioral Disengagement	**−0.42**	**0.39**	**−0.24**	**−0.38**	**−0.36**	**−0.31**	**−0.10**	**0.09**	**−0.17**	**−0.36**	**−0.31**	**0.32**	**0.24**	**−0.21**	0.00	−0.01	**0.21**	-

Variables 5 to 18 are the dimensions measured by the Brief COPE Inventory. Bold font: *p* < 0.0001.

**Table 3 ijerph-19-07112-t003:** Association between variables and quality of life.

	All (*n* = 8469)		NHCI (*n* = 4930)		NPP (*n* = 3078)	
	*β*	95% CI	*β*	95% CI	*β*	95% CI
**Main independent variables**						
Perceived Stress	−0.49 *	[−0.51, −0.48]	−0.49 *	[−0.51, −0.47]	−0.50 *	[−0.52, −0.47]
Social Support	0.21 *	[0.19, 0.22]	0.21 *	[0.19, 0.23]	0.20 *	[0.18, 0.23]
Resilience	0.02	[0.00, 0.04]	0.02	[0.00, 0.05]	0.01	[−0.02, 0.05]
**Coping strategies**						
Active Coping	0.03 *	[0.01, 0.05]	0.04 *	[0.02, 0.07]	0.01	[−0.02, 0.04]
Planning	0.02	[0.00, 0.03]	0.02	[−0.01, 0.04]	0.02	[−0.01, 0.05]
Seeking Instrumental Support	0.02	[0.00, 0.04]	0.02	[0.00, 0.05]	0.02	[−0.01, 0.06]
Seeking Emotional Support	−0.01	[−0.03, 0.01]	0.00	[−0.03, 0.02]	−0.01	[−0.04, 0.02]
Venting	0.04 *	[0.02, 0.06]	0.03 *	[0.01, 0.06]	0.05 *	[0.02, 0.07]
Positive Reframing	0.09 *	[0.08, 0.11]	0.08 *	[0.05, 0.10]	0.12 *	[0.09, 0.15]
Accepance	0.08 *	[0.06, 0.09]	0.08 *	[0.06, 0.11]	0.06 *	[0.03, 0.09]
Denial	−0.04 *	[−0.05, −0.02]	−0.03 *	[−0.05, −0.01]	−0.04 *	[−0.07, −0.02]
Self-Blame	−0.07 *	[−0.09, −0.06]	−0.08 *	[−0.10, −0.06]	−0.06 *	[−0.09, −0.04]
Humor	0.02	[0.00, 0.03]	0.02	[0.00, 0.04]	0.01	[−0.02, 0.04]
Religion	0.02	[0.00, 0.03]	0.01	[−0.01, 0.02]	0.02	[0.00, 0.05]
Self-distraction	0.02	[0.00, 0.03]	0.02	[0.00, 0.04]	0.01	[−0.01, 0.03]
Substance Use	−0.04 *	[−0.05, −0.02]	−0.03 *	[−0.05, −0.01]	−0.04 *	[−0.06, −0.01]
Behavioral disengagement	−0.05 *	[−0.07, −0.04]	−0.05 *	[−0.07, −0.03]	−0.06 *	[−0.09, −0.04]
**Sociodemographic variables**						
Gender: Woman	0.01	[0.00, 0.02]	0.01	[0.00, 0.03]	0.00	[−0.02, 0.02]
Gender: Describes otherwise	0.00	[−0.01, 0.01]	0.00	[−0.01, 0.02]	−0.01	[−0.03, 0.01]
Age: 30–39	−0.07 *	[−0.09, −0.04]	−0.07 *	[−0.09, −0.04]	−0.05	[−0.10, 0.01]
Age: 40–49	−0.10 *	[−0.13, −0.07]	−0.10 *	[−0.13, −0.06]	−0.08	[−0.15, −0.02]
Age: 50+	−0.10 *	[−0.13, −0.07]	−0.10 *	[−0.13, −0.07]	−0.09 *	[−0.15, −0.03]
Marital situation: Married	0.05 *	[0.04, 0.07]	0.06 *	[ 0.04, 0.08]	0.04	[0.01, 0.06]
Marital situation: Other	0.01	[−0.01, 0.02]	0.01	[−0.01, 0.02]	0.01	[−0.01, 0.04]
Having Children: Yes	−0.01	[−0.03, 0.01]	−0.02	[−0.04, 0.00]	0.00	[−0.02, 0.03]
Time since diploma: 5–10 years	0.01	[−0.01, 0.03]	0.02	[0.00, 0.05]	−0.01	[−0.06, 0.03]
Time since diploma: 10+ years	0.04 *	[0.02, 0.06]	0.06 *	[0.03, 0.09]	0.00	[−0.05, 0.05]
**COVID-19-related variables**						
Exposure: Indirect	−0.01	[−0.03, 0.00]	−0.02	[−0.04, 0.01]	−0.01	[−0.04, 0.02]
Exposure: Direct	−0.02	[−0.03, 0.00]	−0.01	[−0.04, 0.01]	−0.03	[−0.06, 0.00]
Reassignment: Yes	0.00	[−0.02, 0.01]	−0.01	[−0.03, 0.01]	−0.01	[−0.04, 0.01]
**Adjusted R^2^**	**0.62**		**0.62**		**0.63**	

*****: *p* < 0.005; Remark: some nurses who participated in the study had a mixed practice alternating between hospital/care institution and private work. They were included in the general analyses, but because of how few they were in number (*n* = 553) compared to the others, they were excluded from the stratified analyses. NHCI: nurses working at hospitals or care institutions; NPP: nurses in private practice; CI: confidence interval.

## Data Availability

The data presented in this study are available on request from the corresponding author. The data are not publicly available due to the presence of health information.

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
