# Peer review of "Protective Factors and Coping Styles Associated with Quality of Life during the COVID-19 Pandemic: A Comparison of Hospital or Care Institution and Private Practice Nurses"

_ijerph, 2022, doi:10.3390/ijerph19127112_

Round 1
Reviewer 1 Report
In the manuscript "Which Protecting Factors and Coping Styles Protected French Nurses' Quality of Life During the COVID-19 Pandemic: Comparison between Hospital or Care Institution and Private Practice Nurses" the authors analyze the protecting effect in the quality of life of French nurses during the COVID-19 pandemic of perceived stress, social support, resilience, coping styles, in addition to sociodemographic variables and of exposure to COVID-19. Furthermore, they analyze the differences in these variables between nurses who work in hospitals or Care institution and in private practice. Also they performed multiple linear regressions with quality of life as outcome, in addition on the whole sample, stratifying by the type of practice
The manuscript is well written and covers important issues. However, some minor changes should be done before publication:
Introduction
-Since it is a central variable of the study, quality of life should be explained more broadly. The definition on page 1, lines 39-41 does not seem sufficient and, moreover, appears in parentheses.
-On page 2, on lines 48 to 50, the following phrase appears “The most impactful risk factors were having direct contact with COVID-19 patients, inadequate protective equipment, heavy workload, fear of infection, and concern for family.” This sentence needs a citation.
Materials and Methods
-On page 2, in lines 86-88 it says that “A link to an online self-administered questionnaire developed with Sphinx iQ2 v7.4.5.1 was sent to all registered French nurses through the Ordre National des Infirmiers followed by three weekly reminders”. The composition of the study sample should also be explained and the response rate obtained should be reported.
-On page 3, line 96 appears “items measuring several aspects of QoL”. It should be mentioned what aspects it refers.
Results
-On page 4, line 159 appears “Bivariate t-test” but results refer to Chi-square test. It should be corrected.
Discussion
In limitations it should be recognized that, although statistically significant, the magnitude of some of the differences between hospital or care institutions (NHCI) and private practice nurses (NPP) is very small (for example, the mean of social support for NHCI was 5.4 (SD = 1.2) and for NPP the mean was 5.3 (SD = 1.3), and also some of the Beta weights from the regression analyzes were very low (for example: -0.04 for Using substances and 0.04 for Time since diploma: 10+years).
Minor errors
-On page 3, on line 143 this text appears “Erreur ! Source du renvoi introuvable.” and the sentence that ends on line 142 is unfinished.
And on page 5, on lines 177 and 178, this text appears: “(Erreur ! Source du renvoi introuvable.)”
Author Response
Reviewer 1 suggested some minor changes at several points of the manuscript. We thank them for their comments and suggestions and have followed all of their advice. Here is the list of their comments (in italics) followed by our responses.
-Since it is a central variable of the study, quality of life should be explained more broadly. The definition on page 1, lines 39-41 does not seem sufficient and, moreover, appears in parentheses.
We developed the paragraph about quality of life et provided more details about what this concept entails (lines 40-48 of the revised manuscript).
-On page 2, on lines 48 to 50, the following phrase appears “The most impactful risk factors were having direct contact with COVID-19 patients, inadequate protective equipment, heavy workload, fear of infection, and concern for family.” This sentence needs a citation.
The sentence in question has disappeared in the rewriting of most of the introduction asked by another reviewer. We tried to include all necessary references in the new text.
-On page 2, in lines 86-88 it says that “A link to an online self-administered questionnaire developed with Sphinx iQ2 v7.4.5.1 was sent to all registered French nurses through the Ordre National des Infirmiers followed by three weekly reminders”. The composition of the study sample should also be explained and the response rate obtained should be reported.
We developed the paragraph about recruitment (lines 102-106) and reported the response rate in the results. The study sample is described at the beginning of the results (lines 168-175) and in Table 1.
-On page 3, line 96 appears “items measuring several aspects of QoL”. It should be mentioned what aspects it refers.
We precised these aspects in the text (lines 44-45).
-On page 4, line 159 appears “Bivariate t-test” but results refer to Chi-square test. It should be corrected.
We clarified that we used both bivariate t-tests and independence Chi2 tests (line 157).
In limitations it should be recognized that, although statistically significant, the magnitude of some of the differences between hospital or care institutions (NHCI) and private practice nurses (NPP) is very small (for example, the mean of social support for NHCI was 5.4 (SD = 1.2) and for NPP the mean was 5.3 (SD = 1.3), and also some of the Beta weights from the regression analyzes were very low (for example: -0.04 for Using substances and 0.04 for Time since diploma: 10+years).
We added this point to the limitations (lines 341-342).
-On page 3, on line 143 this text appears “Erreur ! Source du renvoi introuvable.” and the sentence that ends on line 142 is unfinished.
-And on page 5, on lines 177 and 178, this text appears: “(Erreur ! Source du renvoi introuvable.)”
Thanks for noticing. The links to the Tables somehow broke during files conversion. This has been fixed.
Reviewer 2 Report
Dear Authors,
it is a pleasure to review your work about the nurses' protecting factors related to the quality of life during the COVID pandemic.
Here are my comments that can help you improving the manuscript:
Introduction: you can add some information about the role of resilience and coping strategies in the mental health of HCWs during pandemic, you can check this reference https://www.mdpi.com/1660-4601/18/18/9453
line 143 and following: please provide percentages and numbers; also in the following lines, it makes the reading easier.
is there any difference between NHCI and NPP regarding exposure to COVID patients?
paragraph 3.3. You should provide the results of your analysis, so that the reader can understand the weight of each variable considered associated with QoL. I can't find the difference in the results between NCPI and NPP. Also, I suggest an overall rewriting of the paragraph to be more specific in describing the analysis.
Provide practical implications of your study, especially underlining the differences and similarities between the two groups.
Limitations: please improve this part, an important limitation of your study is that the survey is self adminstered and volounteer, so you miss the non-respondents'characteristics. Also, what was the percentage of answer to the questionnaire? please add in the Results paragraph.
Good luck and best wishes!
Author Response
Reviewer 2 made suggestions to improve the introduction and discussion and asked that we provide further details on some aspects of the methods and results. We thank them for their comments and recommendations and have followed all of their advice. Here is the list of their comments (in italics) followed by our responses.
Introduction: you can add some information about the role of resilience and coping strategies in the mental health of HCWs during pandemic, you can check this reference https://www.mdpi.com/1660-4601/18/18/9453
Thank you for the advice and reference. We included it in the introduction (lines 46-47 and 75-78 in the revised manuscript). Note that most of the Introduction was rewritten based on another reviewer’s comment.
Line 143 and following: please provide percentages and numbers; also in the following lines, it makes the reading easier.
We added all numbers in the text (starting at line 168).
Is there any difference between NHCI and NPP regarding exposure to COVID patients?
We added a comment on this point in the descriptive statistics (189-191). NHCIs were more frequently directly exposed to COVID-19, but they also were more frequently not exposed at all. On the opposite, NPPs were more frequently exposed indirectly. This is might be due to NHCIs often working either in COVID-19-specific units, or in services that never received COVID patients, such as mental health services, whereas NPPs could have been indirectly exposed, for example, when caring at home for patients whose relatives were infected. We did not include this hypothesis in the manuscript because more specific research on the topic is needed to provide evidence for or against it.
Paragraph 3.3. You should provide the results of your analysis, so that the reader can understand the weight of each variable considered associated with QoL. I can't find the difference in the results between NCPI and NPP. Also, I suggest an overall rewriting of the paragraph to be more specific in describing the analysis.
We added standardized regression coefficients in the text to make the associations more clear (lines 222-241). We did not comment on the differences between NHCI and NPP because the associations we observed were almost exactly the same for both groups. We emphasized this point in the discussion (lines 331-334).
Provide practical implications of your study, especially underlining the differences and similarities between the two groups.
We added a paragraph in the discussion about the implications of the study (lines 329-339).
Limitations: please improve this part, an important limitation of your study is that the survey is self adminstered and volounteer, so you miss the non-respondents'characteristics. Also, what was the percentage of answer to the questionnaire? please add in the Results paragraph.
We completed the limitation section (lines 348-349) and provided the response rate to the questionnaire (line 168), which was 2.5%.
Reviewer 3 Report
First of all, I would like to thank you for the opportunity to read your interesting paper entitled “Which Protecting Factors and Coping Styles Protected French Nurses’ Quality of Life During the COVID-19 Pandemic: Comparison between Hospital or Care Institution and Private Practice Nurses”. I think that you are tackling a timely and relevant topic, which deserves attention in the scholarly debate. This is an exciting and well-conceived study of important constructs, i.e., quality of life, stress, resilience, social support, and coping style. The authors used the data of 9898 French nurses working in hospitals, care institutions, and private practitioners. Results revealed that social support and two coping strategies (positive reframing and acceptance) were related to a high quality of life. In contrast, perceived stress and four coping strategies (denial, self-blame, drug use, and behavioral disengagement) were associated with a low quality of life.
Although the paper focuses on essential concepts and their relationship, a few concerns deserve attention. I list here to offer some suggestions for improving the manuscript.
The introduction sounds confusing to me. The authors are unable to identify a major gap in the scientific knowledge they are going to fill in with their research. This prevents them from giving a “shape” to their research, which is presented in a confused and unattractive way in the current version of the manuscript. I recommend the authors carefully rewrite their introduction, trying to: 1) emphasize the gap in the scientific knowledge they are going to fill; 2) stress the relevance of their work and their distinctive contribution to the advancement of scientific literature, and 3) clarify the research questions that are addressed in this research.
It is not really clear why you chose these particular constructs over others. I believe that your contribution would be more valuable if you provided compelling reasons for selecting these constructs over others.
Can you describe in detail how participants were recruited? More information is needed on questionnaire design and sample size selection. Participants and procedures should be elaborated more.
Please add the CFA for all the study variables, compare the measurement model with the alternate models, and then perform convergent and discriminant validity among variables.
Please perform bivariate correlation among all the study variables before regression analysis.
The theoretical and managerial implications part should be incorporated.
As several flaws can be retrieved throughout the manuscript, further proofreading of the paper is warmly recommended.
I would like my recommendations to help the authors improve their work. I hope the authors will benefit from these suggestions and make the necessary amendments to strengthen the manuscript for later submission.
Author Response
Reviewer 3 asked for substantial changes of the introduction aiming at making it clearer, several additional analyses, and made suggestions to improve the discussion. We thank them for their comments and suggestions and have followed most of their advice. Here is the list of their comments (in italics) followed by our responses.
The introduction sounds confusing to me. The authors are unable to identify a major gap in the scientific knowledge they are going to fill in with their research. This prevents them from giving a “shape” to their research, which is presented in a confused and unattractive way in the current version of the manuscript. I recommend the authors carefully rewrite their introduction, trying to: 1) emphasize the gap in the scientific knowledge they are going to fill; 2) stress the relevance of their work and their distinctive contribution to the advancement of scientific literature, and 3) clarify the research questions that are addressed in this research.
It is not really clear why you chose these particular constructs over others. I believe that your contribution would be more valuable if you provided compelling reasons for selecting these constructs over others.
We extensively rewrote the introduction emphasizing the points that were stressed. We hope it is now clearer and provides a better picture of our study's goals.
Can you describe in detail how participants were recruited? More information is needed on questionnaire design and sample size selection. Participants and procedures should be elaborated more.
We provided more details on participants’ recruitment (lines 101-109 of the revised manuscript).
Please add the CFA for all the study variables, compare the measurement model with the alternate models, and then perform convergent and discriminant validity among variables.
As all the scales we used have been validated in English and French (references are provided in the Material and Methods section, lines 115-142), we did not perform these analyses that are not customary for validated scales, go beyond the scope of the current work, and would make the manuscript much longer. Internal consistency of each scale was measured with Cronbach’s alpha that are provided. We clarified in the text that all scales have been validated in English and French (lines 113-114).
Please perform bivariate correlation among all the study variables before regression analysis.
We performed this analysis and added a paragraph summarizing it (lines 200-205) as well as the correlation table in the Results (page 7).
The theoretical and managerial implications part should be incorporated.
We added a paragraph about the implications of the study in the discussion (lines 329-330).
As several flaws can be retrieved throughout the manuscript, further proofreading of the paper is warmly recommended.
We solicited MDPI’s language editing services to have the English professionally checked.
Round 2
Reviewer 2 Report
Dear authors, you properly address my comments, thank you.
Good luck with your work!
Author Response
Reviewer 2 mentioned that we properly addressed their comments.
We thank them again for their clear and constructive remarks that helped make our work as good as possible.